# Effects of socioeconomic factors and booking time on the WHO recommended eight antenatal care contacts in Liberia

**Michael Ekholuenetale**[1], **Chimezie Igwegbe Nzoputam**[2,3], **Amadou Barrow**[4]*

1 Faculty of Public Health, Department of Epidemiology and Medical Statistics, College of Medicine, University of Ibadan, Ibadan, Nigeria, 2 Department of Public Health, Center of Excellence in Reproductive Health Innovation (CERHI), College of Medical Sciences, University of Benin, Benin City, Nigeria, 3 Department of Medical Biochemistry, School of Basic Medical Sciences, University of Benin, Benin City, Nigeria, 4 Department of Public & Environmental Health, School of Medicine & Allied Health Sciences, University of The Gambia, Kanifing, The Gambia

* abarrow@utg.edu.gm

**Data Availability Statement:** Data for this study were sourced and available here: http://dhsprogram.com/data/available-datasets.cfm.

## Abstract

Antenatal care (ANC) is an important intervention that has been linked to reduce maternal and newborn adverse outcomes. However, the long years of war in Liberia may have contributed to the poor health indices including the uptake of maternal health care services. The objective of this study was to determine the marginal interaction effects between booking time and socioeconomic factors in eight or more ANC contacts. A total sample of 4,185 women who had given birth were included in this study. The 2020 Liberia Demographic and Health Survey (LDHS) dataset was analyzed. The outcome variable was eight or more ANC contacts. Percentage and Chi-square test were used in univariate and bivariate analyses respectively. The marginal interaction effects between booking time and socioeconomic factors of eight or more ANC contacts were estimated. The statistical significance was determined at 5%. The weighted prevalence of eight or more ANC contacts was 26.6% (95% CI: 23.8%, 29.6%). The uptake of eight or more ANC contacts increased steadily by increasing women's level of education and household wealth index. Women with higher educational attainment had a prevalence of 49.0% (95%CI: 36.5%, 61.6%) and those in the richest households had an estimated prevalence of 31.4% (95%CI: 24.9%, 38.8%) respectively. Furthermore, the urban dwellers had a weighted eight or more ANC contacts prevalence of 29.0% (95%CI: 24.6%, 34.0%). The key finding is increased marginal interaction effects for higher education and early booking (48.4%), richest households and early booking (35.4%), and urban residential status and early booking (36.2%) respectively. Overall, the prevalence of eight or more ANC contacts was low. However, we found higher coverage of eight or more ANC contacts among women who initiated ANC within the first trimester and among those with higher socioeconomic status. We recommend the Liberian government to design and/or support programmes targeted at promoting early ANC initiation and supporting the disadvantaged women such as the uneducated, poor and those living in rural or remote settings.

**Funding:** The authors received no specific funding for this work.

**Competing interests:** The authors have declared that no competing interests exist.

**Abbreviations:** ANC, Antenatal Care; CI, Confidence Interval; EAs, Enumeration Areas; LDHS, Liberia Demographic and Health Survey; ICF, Inner City Fund; IPTp, Intermittent Preventative Therapy in Pregnancy; MMR, Maternal Mortality Ratio; PCA, Principal Components Analysis; SDGs, Sustainable Development Goals; SE, Standard Error; TX, Texas; UHC, Universal Health Coverage; USA, United States of America; WHO, World Health Organization.

## Introduction

Pregnancy-related problems account for more than half of all women deaths in many resource-poor settings every year. World Health Organization (WHO) reported that 9 of every 10 maternal death are recorded in resource-constrained settings [1]. The report of maternal deaths have prompted a global call for immediate action, as the majority of these deaths can be avoided with skilled birth attendance and increased antenatal care (ANC) contacts [2]. In 2016, WHO made a recommendation to increase the frequency of ANC contacts with skilled medical personnel, from four to eight [3–5]. This was seen as a potential strategy to reduce maternal deaths by up to 8 per 1000 live births [6]. This recommendation is believed to aid the implementation of preventive measures, especially early detection of danger signs, reduction of complications, and the elimination of health care inequities, particularly in the rural and marginalized areas [7]. The guideline could increase the chance to improve maternal health services uptake including; dietary counseling, blood test, use of intermittent preventative therapy in pregnancy (IPTp) and the tetanus immunization [6].

The staggering maternal deaths is disproportionately higher among women of low socioeconomic class [1,8,9]. The disadvantaged women and their newborns continue to die due to complications during pregnancy, childbirth or the weeks following delivery (puerperium), especially in developing countries, such as sub-Saharan Africa (SSA) and southern Asia. The global maternal mortality ratio (MMR) declined from 385 deaths per 100,000 livebirths in 1990, to 216 in 2015, corresponding to a relative decline of 43·9%, with 303,000 maternal deaths worldwide [8]. Meanwhile, the regional MMRs was 546 deaths per 100,000 livebirths for SSA [8]. This is worrisome and requires accelerated progress to achieve the Sustainable Development Goals (SDGs) targeted at reducing maternal mortality [10].

It is difficult to estimate the socioeconomic burden due to maternal death especially in resource-constrained settings [11]. Several factors have been identified to be associated with maternal deaths. For example, inadequate access to maternal health care services, increases the risk of pregnancy and obstetric issues. The uptake of adequate ANC contacts is influenced by several factors such as socioeconomic factors [12], maternal age [7], educational level [13], intended pregnancies [14], women's enlightenment and timing of first ANC visit [2]. In addition, exposure to media, family income, and obstetric service accessibility have been linked to higher prenatal care service uptake [14,15].

There has been a growing interest in the supply and accessibility of maternal health care services in developing countries, particularly those that have recently emerged from long years of civil conflict, such as Liberia. The goal is to undertake concerted efforts to reduce maternal and newborn mortality, which has continued to be a major problem [8], especially that disparities in the distribution of maternal health care services exist in Liberia. For example, about 90% of women in the highest income group had a delivery attended by skilled personnel as compared to 43% of women in the lowest income group [16]. Liberia's MMR is among the highest in the world. The key attributable factor is inadequate ANC visit [16].

The prolonged civil war that ravaged the country came to an end in 2003. During the conflict, the healthcare system was ravaged, with the majority of healthcare institutions devastated, thereby leaving the country with some of the worst health indices in the world. Due to the massive collapse of healthcare system, women became more reliant on alternative healthcare services, such as traditional healers [17]. The poor healthcare system [18] is a notable effect of the two Liberian civil wars (between 1989 and 1996 and 1997–2003 respectively). After years of civil war and the worst Ebola outbreak in history, the Liberian health system has struggled to rebuild a comprehensive maternal health care.

Liberia is currently ranked as one of the countries with the highest MMR worldwide, with a staggering record of approximately 725 deaths per 100,000 live births [19]. Currently, Liberia is facing supply-side factors such as inadequate infrastructure, lack of competent and sufficient health care providers and medical equipment [20]. However, the health sector is making efforts to revamp the healthcare system and improve women's access to health care services. Since the end of the civil unrest, the Liberian government has begun a large-scale effort to reconstruct the healthcare system like some other developing countries [21]. By selecting high-priority services, a cost-effective and evidence-based intervention to improve maternal health has been linked to increase access to health care and lower morbidity and mortality rates [22]. To our best of knowledge, no study has been found in Liberia that looked at the coverage and factors associated with the revised WHO ANC guideline of eight ANC contacts. The aim of this study was to investigate the marginal interaction effect between ANC booking time and socioeconomic factors in eight ANC contacts in Liberia.

## Methods

### Ethics statement

We used secondary data available in the public domain. The procedure for the 2019/20 LDHS has been approved by the ICF Institutional Review Board and the Liberia Health Service Ethical Review Committee. Participants have been informed of the benefits and risks of participating in the survey. Written informed consent was obtained directly from qualified respondents prior to administration of the Household or Women's Questionnaire. The survey was entirely voluntary. The final datasets did not include the respondents' names or identification numbers. No further approval was required for this study. More details about data and ethical standards are available at http://goo.gl/ny8T6X.

### Data source

The datasets used in this study were obtained from the Liberia Demographic and Health Survey for 2019/2020 (LDHS). The individual women dataset, which included large representative samples of women aged 15–49 years, was used in the analysis. This study examined data from 4185 women. To obtain a nationally representative sample of respondents, a questionnaire for individual women aged 15–49 was used. USAID commissioned the population-based surveys, which were carried out by the Liberian government with operational assistance from ICF International. The survey use a two-stage stratified cluster design. Clusters or Enumeration Areas (EA) were first chosen from a population sample frame, and households were then systematically chosen from the clusters. The MEASURE DHS programme developed standardized questionnaires for households, men, and women, which were administered during face-to-face interviews. The details of DHS data has been report in a previous study [23].

### Variable selection and measurement

**Outcome.** The frequency of ANC contacts with physicians, nurses, and midwives was measured on a dichotomous scale. The LDHS asked the question *"Number of antenatal visits during pregnancy*?" This question's responses were classified as $< 8$ or $\geq 8$ contacts. The WHO ANC guideline recommendations map to the eight suggested contacts and provide a review structure for the 2016 WHO ANC [3–5].

**Socioeconomic variables.** Women's educational attainment, household wealth quintiles, residential and employment status were used to assess socioeconomic status in accordance with previous research [24–26]. Women's educational attainment was classified as follows: no

formal education, primary, secondary, or higher. The location of residence: urban vs. rural. Women's employment was classified as either employed or unemployed. The wealth indicator weights were assigned using the principal component analysis (PCA) technique. Wealth indicator variable scores were assigned and standardized based on household assets such as wall type, floor type, roof type, water supply, sanitation facilities, radio, electricity, television, refrigerator, cooking fuel, furniture, and room occupancy. The factor loadings and z-scores were then calculated. The indicator values were multiplied by the factor loadings for each household and summarized to produce the household's wealth index value. The standardized z-score was used to divide the overall scores into wealth quintiles: poorest, poorer, middle, richest, and richest [27].

**Explanatory variables.**   The independent variables include: total children ever born: 1–2, 3–4, 5+; health insurance coverage: covered vs. not covered; religion: Christianity, Islam and traditional/others; region: North Western, South Central, South Eastern A, South Eastern B and North Central; marital status: single, currently married/in union, formerly married; age: 15–19, 20–24, 25–29, 30–34, 35–39, 40–44, 45–49; wanted child when became pregnant: then, later, no more; ANC booking time: late (after 1st trimester), early (within 1st trimester).

## Analytical approach

The survey module ('svy') command was used to modify the sampling design. A variance inflation factor of 10 was used to determine multicollinearity, which is known to be a major source of concern in regression models [28,29]. However, no variable was removed from the model because it was determined that they were not interdependent. Percentage was used in univariate analysis. The Chi-square test was used to investigate the relationship between eight ANC contacts and the explanatory variables. Statistical significance was determined at p< 0.05. Stata version 14 (StataCorp., College Station, TX, USA) was used for data analysis.

## Marginal effect modelling

A marginal effect is the change in the predicted value of eight or more ANC contacts after changing an explanatory factor, either a discrete change in categorical variables or an instantaneous change in continuous variables, while all other variables are held at specified values. The predictive marginal effect model included all significant variables from the bivariate analysis (with corresponding 95% CI). The predictive marginal effect model is presented thus;

$$\Pr(Y = 1 \mid \mathrm{Set}[\mathrm{E} = e]) = \sum_z \hat{p}_{ez} \Pr(Z = z);$$

Where Set[E = e] reflects putting all observations to a single exposure level e, and Z = z refers to a given set of observed values for the covariate vector Z. Furthermore, $\hat{p}_{ez}$ is the predicted probabilitis of $\geq$ 8 ANC contacts for any E = e and Z = z. The marginal effects indicate a weighted average over the distribution of the covariates and are equal to estimates got by standardizing to the entire population. As a post logit test, the exposure *E* is set to the level *e* for all women in the dataset, and the logit coefficients are used to compute predicted probabilities for every woman at their observed covariate pattern and newly exposure value. Because predicted probabilities are computed under the same distribution of *Z*, there is no covariate of the corresponding effect measure estimates [30,31].

## Results

The weighted prevalence of eight or more ANC contacts was 26.6% (95% CI: 23.8%, 29.6%). In Table 1, the uptake of eight or more ANC contacts increased steadily by increasing women's

**Table 1.** Distribution of eight or more ANC contacts in Liberia (n = 4,185).

| Variable | n (%) | Prevalence of eight or more ANC contacts | P |
|---|---|---|---|
| **Socioeconomic variables** | | | |
| **Education** | | | <0.001* |
| No formal education | 1688 (40.3) | 24.2 (21.5, 27.2) | |
| Primary | 1231 (29.4) | 24.4 (21.0, 28.1) | |
| Secondary | 1181 (28.2) | 27.6 (23.6, 32.0) | |
| Higher | 85 (2.0) | 49.0 (36.5, 61.6) | |
| **Household wealth quintile** | | | 0.042* |
| Poorest | 1109 (26.5) | 24.8 (20.9, 29.1) | |
| Poorer | 914 (21.8) | 22.3 (19.1, 25.8) | |
| Middle | 825 (19.7) | 24.8 (19.4, 31.0) | |
| Richer | 712 (17.0) | 30.7 (25.4, 36.6) | |
| Richest | 625 (14.9) | 31.4 (24.9, 38.8) | |
| **Residential status** | | | 0.033* |
| Urban | 1493 (35.7) | 29.0 (24.6, 34.0) | |
| Rural | 2692 (64.3) | 23.4 (21.0, 26.0) | |
| **Employment** | | | 0.233 |
| Employed | 2790 (66.7) | 24.9 (21.0, 29.2) | |
| Unemployed | 1395 (33.3) | 27.5 (24.4, 30.6) | |
| **Other women's characteristics** | | | |
| **Total children ever born** | | | 0.018* |
| 1–2 | 1726 (41.2) | 26.7 (22.8, 31.0) | |
| 3–4 | 1092 (26.1) | 30.3 (26.3, 34.7) | |
| 5+ | 1367 (32.7) | 22.7 (19.4, 26.3) | |
| **Health insurance coverage** | | | 0.479 |
| Covered | 128 (3.1) | 31.4 (18.5, 48.0) | |
| Not covered | 4057 (96.9) | 26.5 (23.7, 29.4) | |
| **Religion** | | | 0.160 |
| Christianity | 3556 (85.0) | 26.0 (23.0, 29.1) | |
| Islam | 541 (12.9) | 31.4 (24.8, 38.9) | |
| Traditional/others | 88 (2.1) | 18.2 (8.2, 35.7) | |
| **Region** | | | 0.530 |
| North Western | 626 (15.0) | 24.6 (20.3, 30.0) | |
| South Central | 1098 (26.2) | 28.1 (22.8, 34.1) | |
| South Eastern A | 630 (15.1) | 28.2 (23.1, 33.9) | |
| South Eastern B | 738 (17.6) | 26.1 (22.0, 30.7) | |
| North Central | 1093 (26.1) | 24.9 (21.6, 28.5) | |
| **Marital status** | | | 0.034* |
| Single | 888 (21.2) | 22.8 (17.7, 28.9) | |
| Currently married/in union | 2910 (69.5) | 28.7 (25.8, 31.7) | |
| Formerly married | 387 (9.3) | 22.1 (16.8, 28.4) | |
| **Age** | | | 0.011* |
| 15–19 | 470 (11.2) | 20.2 (15.1, 26.6) | |
| 20–24 | 971 (23.2) | 27.8 (23.5, 32.6) | |
| 25–29 | 897 (21.4) | 26.9 (22.4, 32.0) | |
| 30–34 | 690 (16.5) | 32.4 (27.2, 38.1) | |
| 35–39 | 665 (15.9) | 24.5 (19.5, 30.3) | |
| 40–44 | 356 (8.5) | 25.8 (19.6, 33.1) | |

*(Continued)*

**Table 1.** (Continued)

| Variable | n (%) | Prevalence of eight or more ANC contacts | P |
|---|---|---|---|
| 45–49 | 136 (3.3) | 13.2 (7.6, 21.9) | |
| **Wanted child when became pregnant** | | | <0.001* |
| Then | 2429 (58.0) | 30.7 (27.4, 34.3) | |
| Later | 1384 (33.1) | 22.6 (19.1, 26.5) | |
| No more | 372 (8.9) | 15.2 (11.1, 20.4) | |
| **ANC Booking time** | | | <0.001* |
| Early (within 1st trimester) | 2955 (72.2) | 34.7 (31.3, 38.1) | |
| Late (after 1st trimester) | 1140 (27.8) | 7.4 (5.0, 10.6) | |

*Significant at p<0.05.

level of education and household wealth quintile; with women in higher educational attainment having a prevalence of 49.0% (95%CI: 36.5%, 61.6%) and those in the richest households having an estimated prevalence of 31.4% (95%CI: 24.9%, 38.8%) respectively. Furthermore, the urban dwellers had a weighted eight or more ANC contacts prevalence of 29.0% (95%CI: 24.6%, 34.0%). In addition, those who booked early (initiated ANC contacts within 1st trimester) had a prevalence of 34.7% (95%CI: 31.3%, 38.1%). Women who planned for pregnancy and those currently married/in union had the highest eight or more ANC contacts prevalence of 30.7% (95%CI: 27.4%, 34.3%). See Table 1 below for the details.

In Table 2, we presented the predictive marginal interaction effect between booking time and education, household wealth and residential status in eight or more ANC contacts. The marginal predictive analysis was conducted to decipher the effects of socioeconomic factors with eight or more ANC contacts while adjusting for other maternal characteristics. From the predictive marginal effects results, assuming that the distribution of all factors remained the

**Table 2. Marginal interaction effect of socioeconomic factors and eight or more ANC contacts in Liberia.**

| Variable | Marginal effect | 95% Confidence Interval | P |
|---|---|---|---|
| **Socioeconomic variables** | | | |
| **Education** | | | |
| No formal education | 25.3 | 21.7, 28.8 | <0.001* |
| Primary | 25.7 | 21.6, 29.8 | <0.001* |
| Secondary | 27.6 | 24.2, 31.0 | <0.001* |
| Higher | 42.3 | 30.0, 54.6 | <0.001* |
| **Booking time** | | | |
| Early (within 1st trimester) | 34.2 | 30.9, 37.5 | <0.001* |
| Late (after 1st trimester) | 8.0 | 5.0, 11.0 | <0.001* |
| **Education-booking time interaction** | | | |
| No formal education*late booking | 5.9 | 2.0, 9.8 | 0.003* |
| No formal education*early booking | 32.5 | 28.2, 36.9 | <0.001* |
| Primary*late booking | 8.9 | 2.1, 11.7 | 0.005* |
| Primary*early booking | 32.8 | 27.6, 38.0 | <0.001* |
| Secondary*late booking | 8.1 | 4.0, 12.1 | <0.001* |
| Secondary*early booking | 34.9 | 30.6, 39.2 | <0.001* |
| Higher*late booking | 26.3 | 1.4, 51.2 | <0.001* |
| Higher*early booking | 48.4 | 34.6, 62.7 | <0.001* |
| **Household wealth quintile** | | | |

(Continued)

**Table 2.** (Continued)

| Variable | Marginal effect | 95% Confidence Interval | P |
|---|---|---|---|
| Poorest | 27.5 | 23.2, 31.7 | <0.001* |
| Poorer | 23.8 | 20.6, 26.9 | <0.001* |
| Middle | 26.3 | 20.7, 31.9 | <0.001* |
| Richer | 29.5 | 24.3, 34.7 | <0.001* |
| Richest | 28.2 | 21.8, 34.5 | <0.001* |
| **Household wealth quintile-booking time interaction** | | | |
| Poorest*late booking | 7.9 | 3.3, 12.5 | 0.001* |
| Poorest*early booking | 34.9 | 29.5, 40.3 | <0.001* |
| Poorer*late booking | 4.9 | 1.6, 8.1 | 0.003* |
| Poorer*early booking | 30.8 | 26.5, 35.1 | <0.001* |
| Middle*late booking | 7.2 | 1.9, 12.4 | 0.007* |
| Middle*early booking | 33.5 | 26.0, 40.9 | <0.001* |
| Richer*late booking | 10.9 | 4.2, 17.6 | 0.001* |
| Richer*early booking | 36.5 | 30.5, 42.6 | <0.001* |
| Richest*late booking | 8.9 | -0.01, 18.6 | 0.076 |
| Richest*early booking | 35.4 | 28.6, 42.3 | <0.001* |
| **Residential status** | | | |
| Urban | 28.6 | 24.1, 33.1 | <0.001* |
| Rural | 24.9 | 22.3, 27.6 | <0.001* |
| **Residential status-booking time interaction** | | | |
| Urban*late booking | 8.6 | 4.3, 12.8 | <0.001* |
| Urban*early booking | 36.2 | 30.7, 41.7 | <0.001* |
| Rural*late booking | 7.2 | 2.9, 11.6 | 0.001* |
| Rural*early booking | 31.6 | 28.4, 34.8 | <0.001* |
| **Other women's characteristics** | | | |
| **Total children ever born** | | | |
| 1–2 | 26.6 | 22.8, 30.4 | <0.001* |
| 3–4 | 28.5 | 24.5, 32.5 | <0.001* |
| 5+ | 26.2 | 20.7, 31.7 | <0.001* |
| **Marital status** | | | |
| Single | 23.1 | 18.2, 27.9 | <0.001* |
| Currently married/in union | 29.0 | 25.9, 32.1 | <0.001* |
| Formerly married | 24.4 | 18.4, 30.4 | <0.001* |
| **Age** | | | |
| 15–19 | 26.9 | 18.9, 34.9 | <0.001* |
| 20–24 | 29.5 | 24.5, 34.4 | <0.001* |
| 25–29 | 25.7 | 21.4, 30.1 | <0.001* |
| 30–34 | 29.0 | 24.1, 33.9 | <0.001* |
| 35–39 | 24.2 | 18.8, 29.6 | <0.001* |
| 40–44 | 28.4 | 21.3, 35.6 | <0.001* |
| 45–49 | 16.1 | 8.1, 24.2 | <0.001* |
| **Wanted child when became pregnant** | | | |
| Then | 29.3 | 26.1, 32.5 | <0.001* |
| Later | 24.6 | 20.7, 28.5 | <0.001* |
| No more | 19.8 | 14.2, 25.4 | <0.001* |

*Significant at p<0.05.

same, but every woman had higher education, we would expect 42.3% of eight or more ANC contacts. If every woman booked early (within 1$^{st}$ trimester), we would expect 34.2% of eight or more ANC contacts. If instead, every women had higher education and booked late (after 1$^{st}$ trimester), we would expect 26.3% of eight or more ANC contacts. Moreover, if every woman had higher education and booked early (within 1$^{st}$ trimester), we would expect 48.4% of eight or more ANC contacts. By implication, we found increased marginal interaction effect between higher education and early booking than higher education and late booking. If instead, the distribution of other factors remained the same among women, but every woman is in the richest household and booked early (within 1$^{st}$ trimester), we would expect 35.4% of eight or more ANC contacts. If every woman is in the richest household and booked late (after 1$^{st}$ trimester), we would expect about 8.9% of eight or more ANC contacts. Furthermore, if instead, the spread of the aforementioned variables was as observed while all covariates remain equal, but every woman lived in the urban residence and booked early (within 1$^{st}$ trimester), we would expect 36.2% of eight or more ANC contacts, but if all woman lived in the urban residence and booked late, we would expect 8.6% of eight or more ANC contacts. The key finding is increased marginal interaction effects for higher education and early booking (48.4%), richest households and early booking (35.4%), and urban residential status and early booking (36.2%) respectively. The details of the predictive marginal interaction effects of the factors associated with eight or more ANC are shown in Table 2 below.

Fig 1 showed the marginal effects plot between booking time and educational attainment in eight or more ANC contacts. The marginal interaction effects in eight or more ANC contacts among the Liberian women was higher for women who booked early than those who booked late. Notably, the marginal interaction effects in eight or more ANC contacts was higher among women who booked early (within 1$^{st}$ trimester) across various educational levels (higher, secondary, primary and without formal education).

In Fig 2, we presented the marginal effects plot between booking time and household wealth in eight or more ANC contacts. The marginal interaction effects in eight or more ANC contacts, was higher for women who booked early than those who booked late. In addition, the marginal interaction effects in eight or more ANC contacts was higher among women who

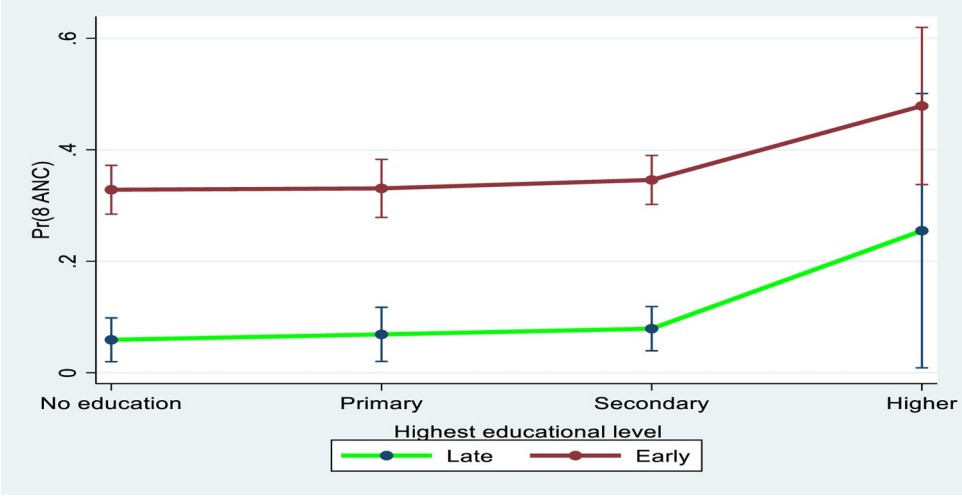

**Fig 1. Predictive marginal effects between early booking and educational attainment in eight or more ANC contacts.**

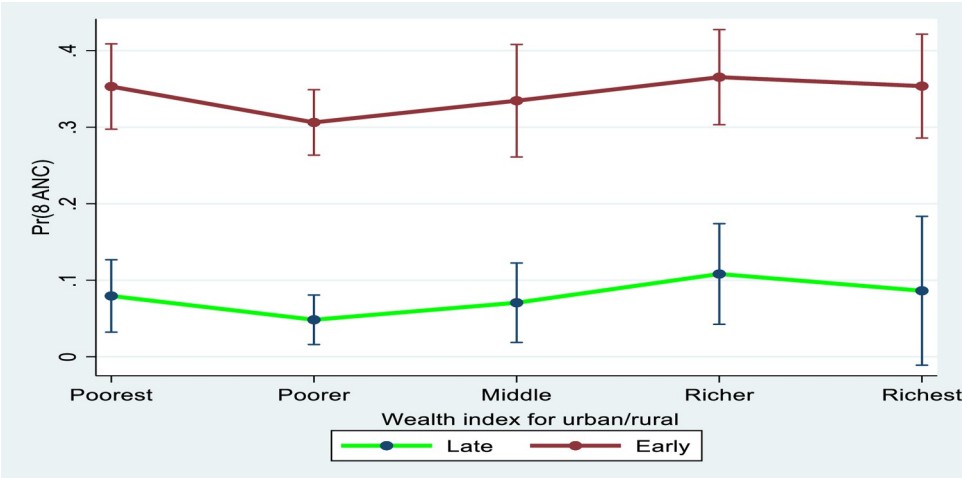

**Fig 2. Predictive marginal effects between early booking and household wealth quintile in eight or more ANC contacts.**

booked early (within 1st trimester) across household wealth quintiles (poorest, poorer, middle, richer and richest).

Fig 3 revealed the marginal effects plot between booking time and residential status in eight or more ANC contacts. The marginal interaction effects in eight or more ANC contacts was higher for women who booked early than those who booked late. Furthermore, the marginal interaction effects in eight or more ANC contacts was higher among women who booked early (within 1st trimester) in urban and rural residence.

## Discussion

This is the foremost study to assess the recent WHO ANC guideline using the 2020 LDHS. Approximately one-quarters of women had eight or more ANC contacts. This prevalence is grossly inadequate considering the maternal mortality burden in Liberia. The prevalence of eight or more ANC contacts in this study is higher than the previous report in some other

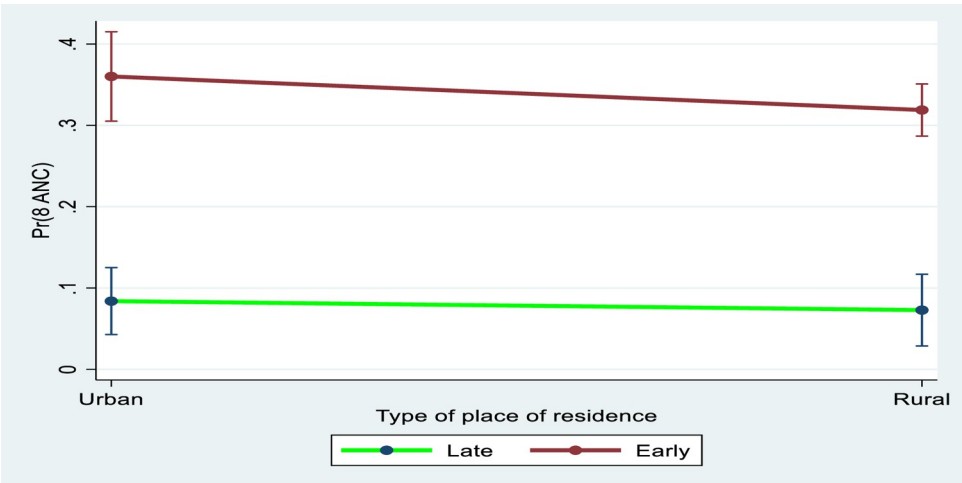

**Fig 3. Predictive marginal effects between early booking and residential status in eight or more ANC contacts.**

countries, such as Senegal (1.0%), Uganda (1.0%), Zambia (1.0%), Mozambique (2.0%), Mali (3.0%), Guinea (3.0%), Cameroon (8.0%), Benin (9.0%), Papua New Guinea (10.0%), Pakistan (14.0%), and Nigeria (20.0%) [32]. Few other countries, such as Albania (30.0%), Ghana (44.0%), and Jordan (74.0%) showed a greater prevalence of eight or more ANC contacts [32]. The low prevalence reported from the previous multi-country study was expected in our study given that the majority of the study countries were low- and middle-income, where health care services are commonly found to be inadequate, substandard or non-existent.

Also, the economic development index of Liberia is similar to that of other resource-constrained settings. The availability and uptake of mother and child health care services in low- and middle-income countries are inadequate due to a lack of infrastructure, poor levels of education, low enlightenment, poverty, even when services are accessible, women are forced to pay out-of-pocket which hinders many in the uptake. To improve maternal health care service utilization, improving the infrastructure, policy formation in the areas of maternal and child health, women empowerments and comprehensive health insurance coverage, particularly for reproductive-age women are required.

More specifically, our research found that higher education and early antenatal bookings had a greater marginal effect in eight or more ANC contacts. When compared with women with lower educational attainment who booked early, those with higher educational attainment and who booked early had greater eight or more ANC uptake. This means that educational attainment and early ANC booking have a positive interaction effect. From the least educated to the most educated, there was a considerable positive interaction effect with early booking. The interaction effects clearly demonstrate that scheduling ANC within the first trimester, will enhance women's uptake of eight or more ANC contacts. The implication is that early ANC booking and improvement in maternal education will help to achieve the WHO's recommended eight or more ANC contacts. From previous studies [33–35], education and enlightenment are major positive factors that enhance women's uptake of maternal and child care services. Other studies have also validated the assertion that women with higher educational attainment are more likely to use maternal health care services, when compared with their counterparts [36–39]. To improve the level of knowledge or enlightenment, girl-child education should be given utmost importance. Women should be exposed to educational materials and other awareness campaign channels such as watching television and listening to health information on radio amongst other behaviour change communication media. No doubt, exposure to health information will contribute to women's knowledge of the importance of early booking and completion of eight or more ANC contacts for a successful pregnancy.

The interaction between household wealth and booking time in eight or more ANC contacts was also investigated. The results showed increased marginal interaction effects between the wealthiest households and early bookings in eight or more ANC contacts. By implication, women who come from the most affluent households and book early will have a greater uptake of ANC contacts. The uptake of ANC and other maternal care services has been positively impacted by wealth status especially in countries such as Liberia where less than 5% of the women are reportedly covered by health insurance. According to a recent study, women from wealthy families are more likely to access and use maternal health care services, when compared with women from low income families [40]. Moreover, women from wealthy households are able to afford health care services especially in out-of-pocket health expenses situation [41]. Besides intervention to promote early booking, improving women's wealth status should also be a priority. To address the issues surrounding poverty in low uptake of health care services, the concept of Universal Health Coverage (UHC) was launched to promote health and well-being and to extend life expectancy for all through access to quality health care

[42]. This assumes that no one is to be left behind. Achieving UHC entails providing everyone with complete access to high-quality healthcare services, as well as financial risk protection.

Promoting health insurance coverage in Liberia will be a mechanism for increasing ANC contacts. At the time of ANC service, there should be no deductibles, copayments, coinsurance payments, or other additional payments. Out-of-pocket expenses could be a significant impediment to making at least eight ANC contacts. A study comparing health care expenditures across SSA found that the majority of countries reported catastrophic health expenditures, with out-of-pocket expenses ranging from one-fifth to more than two-thirds of total health expenditures, indicating that health care remains largely inaccessible despite the introduction of health insurance schemes [43]. This could also be the situation.

Also, we found a significant positive marginal interaction effect between urban residence and early bookings in eight or more ANC contacts. Women who lived in the urban area and booked early had higher coverage for eight or more ANC contacts, when compared with those who lived in rural area. This suggests a link between living in urban residence and booking early in eight or more ANC contacts. Previous studies also found urban dwellers had more ANC contacts than their rural counterparts [44,45]. This urban-rural inequalities in maternal health care service uptake is a major concern. This finding is common in many resource-constrained settings due to disparity in the distribution of functional health facilities which is usually in favour of the urban residence. In addition to promoting early booking, health facilities in rural, remote and hard-to-reach neighbourhoods should be revamped with infrastructure, health care personnel and comprehensive services as commonly available in the urban residence.

To improve optimal uptake of ANC contacts, it is important to design and implement comprehensive maternal health programs and policies. The illiterate, poor and women who live in hard-to-reach communities should be targeted for improved ANC service use. The provision of maternal health care services and health insurance coverage have been recommended [44,46]. Out-of-pocket payments for maternal health services will remain a major barrier to health care access [33], except the Liberian government rise up to this challenge. Based on the findings of this study, if a woman lacks formal education, poor, or lives in a rural residence, early booking alone will not be sufficient to obtain the eight or more ANC contacts.

## Strength and limitation

A significant strength of this study was the use of nationally representative high-quality data from a household survey. Furthermore, appropriate statistical adjustments for the survey designs ensure that the results of this study are reliable. Notwithstanding, a significant limitation of this paper is the possibility of recall bias, which could result in an over- or under-estimation of eight or more ANC contacts. LDHS also does not collect household income or expenditure data, which are the traditional metrics for calculating wealth. The asset-based wealth index used here is merely a proxy indicator of household economic status, and it does not always produce results that are comparable to those obtained from direct income and expenditure measurements when such data is available or can be collected accurately.

## Conclusion

There was low uptake of eight or more ANC contacts in Liberia, as only about one-quarters of women had met the recommended frequency of ANC contacts. We found higher coverage of eight ANC visits among women who had early ANC initiation (within the first trimester) and among those with higher socioeconomic status. We recommend the Liberian government to design and/or support programmes targeted at promoting early ANC initiation (i.e within first

trimester) and supporting the disadvantaged women such as the uneducated, poor and those living in rural or remote settings.

## Acknowledgments

The authors appreciate the Demographic and Health Survey for the approval and access to the original data.

## Author Contributions

**Conceptualization:** Michael Ekholuenetale, Chimezie Igwegbe Nzoputam, Amadou Barrow.

**Data curation:** Michael Ekholuenetale, Chimezie Igwegbe Nzoputam, Amadou Barrow.

**Formal analysis:** Michael Ekholuenetale, Chimezie Igwegbe Nzoputam, Amadou Barrow.

**Methodology:** Michael Ekholuenetale, Chimezie Igwegbe Nzoputam, Amadou Barrow.

**Software:** Michael Ekholuenetale.

**Supervision:** Michael Ekholuenetale, Chimezie Igwegbe Nzoputam.

**Validation:** Michael Ekholuenetale, Chimezie Igwegbe Nzoputam, Amadou Barrow.

**Visualization:** Michael Ekholuenetale, Chimezie Igwegbe Nzoputam, Amadou Barrow.

**Writing – original draft:** Michael Ekholuenetale, Chimezie Igwegbe Nzoputam, Amadou Barrow.

**Writing – review & editing:** Michael Ekholuenetale, Chimezie Igwegbe Nzoputam, Amadou Barrow.

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
