## [Decision Letter · Decision Letter 0]

3 Dec 2021

PGPH-D-21-00801

Effects of socioeconomic factors and booking time on the WHO recommended eight antenatal care contacts in Liberia

Dear Dr. Barrow,

Thank you for submitting your manuscript to PLOS Global Public Health. After careful consideration, we feel that it has merit but does not fully meet PLOS Global Public Health’s publication criteria as it currently stands. Therefore, we invite you to submit a revised version of the manuscript that addresses the points raised during the review process.

We look forward to receiving your revised manuscript.

Kind regards,

Vanessa Moraes Bezerra

Academic Editor

Journal Requirements:

1. Please provide separate figure files in .tif or .eps format only, and remove any figures embedded in your manuscript file.  If you are using LaTeX, you do not need to remove embedded figures.

Additional Editor Comments (if provided):

Work with a high theme for public health. After reading the article and also the opinion described by the reviewer, it is suggested that the author can make adjustments and resubmit the work to be appreciated by the journal.

Reviewers' comments:

Reviewer's Responses to Questions

**Comments to the Author**

1. Does this manuscript meet PLOS Global Public Health’s publication criteria? Is the manuscript technically sound, and do the data support the conclusions? The manuscript must describe methodologically and ethically rigorous research with conclusions that are appropriately drawn based on the data presented.

Reviewer #1: Partly

2. Has the statistical analysis been performed appropriately and rigorously?

Reviewer #1: No

3. Have the authors made all data underlying the findings in their manuscript fully available (please refer to the Data Availability Statement at the start of the manuscript PDF file)?

Reviewer #1: Yes

4. Is the manuscript presented in an intelligible fashion and written in standard English?

Reviewer #1: Yes

5. Review Comments to the Author

Reviewer #1: This study aimed to investigate the marginal interaction effect between ANC booking time and socioeconomic factors in eight ANC contacts in Liberia. The introduction delimits the theme well, exploring the changes in the WHO recommendations for the reduction of maternal deaths. Finally, it highlights the high maternal mortality rates in Sub-Saharan Africa, South Asia and Liberia. This is a study with secondary data from the Liberia Demographic and Health Survey for 2019/2020 (LDHS) that used the number of prenatal visits during pregnancy (<8 or ≥8 contacts) as an outcome. Wealth indicator weights were assigned using the principal component analysis (PCA) technique. Independent variables include: total number of children born; health insurance; religion; region; marital status; age; she wanted a child when she got pregnant; ANC booking time. The Chi-square test was used to investigate the relationship between eight ANC contacts and the Explanatory variables. The predictive marginal effect model included all significant variables from the bivariate analysis (with the corresponding 95% CI). However, the choice of the predictive marginal effect model is not well described in the Methods, especially knowing that the use of these models is more suitable for binary variables with responses (0.1). The sample is large enough to estimate marginal effects for standardization for the entire population, and the results presented in Table 2 are revealing. The graphs highlight the differences between education, family wealth and housing. As a suggestion, authors can remove the colored lines, because line graph is more suited to visualizing trends and movements over time numerically and evenly distributed. Critical points for review: The choice of the predictive marginal effect model is not well described in the Method section. The results, in part, were discussed with the scientific literature and some limitations pointed out. The conclusion does not primarily respond to the objective of the study, and must be rewritten, taking advantage of the findings in table 2. I do not recommend the publication of the manuscript in its current format, but the study itself shows sufficient potential for authors to be encouraged to resubmit a revised version.

6. PLOS authors have the option to publish the peer review history of their article (what does this mean?). If published, this will include your full peer review and any attached files.

**Do you want your identity to be public for this peer review?** For information about this choice, including consent withdrawal, please see our Privacy Policy.

Reviewer #1: No

---

## [Editor Report · Decision Letter 1]

1 Feb 2022

Effects of socioeconomic factors and booking time on the WHO recommended eight antenatal care contacts in Liberia

PGPH-D-21-00801R1

Dear Mr. Barrow,

We are pleased to inform you that your manuscript 'Effects of socioeconomic factors and booking time on the WHO recommended eight antenatal care contacts in Liberia' has been provisionally accepted for publication in PLOS Global Public Health.

Best regards,

Vanessa Moraes Bezerra

Academic Editor

The authors complied with all recommendations in the opinion and the revised version of the manuscript is suitable for publication in the journal.